# A Descriptive Study on the Carcass, Muscle, and Offal Yields of the Plains Zebra (*Equus quagga*) Harvested in Two Seasons

**DOI:** 10.3390/ani13010050

**Published:** 2022-12-22

**Authors:** Angelique Myburgh, Helet Lambrechts, Louwrens Christiaan Hoffman

**Affiliations:** 1Department of Animal Sciences, University of Stellenbosch, Private Bag X1, Matieland, Stellenbosch 7602, South Africa; 2Center for Nutrition and Food Sciences, Queensland Alliance for Agriculture and Food Innovation (QAAFI), The University of Queensland, Digital Agricultural Building, 8115, Office 110, Gatton, QLD 4343, Australia

**Keywords:** game meat, carcass yields, offal yields, muscle yields, zebra

## Abstract

**Simple Summary:**

Equine species are not a common protein source in South Africa, despite being popular in European countries. Horses are known for producing highly nutrient-dense, lean meat, with high dressing percentages. An African equine species with promising production potential is the Plains zebra, which has received very little attention in this regard. The Plains zebra is known to produce nutritious lean meat, but little is known about the carcass composition. Due to its natural resistance to foot-and-mouth disease and the lack of current meat export restrictions for African Horse Sickness, Plains zebra meat is a game meat that can thus be exported from South Africa. This makes it an ideal species for meat production, with the game meat industry being heavily focused on growth, expansion, and formalization. Therefore, the aim of this study was to determine the meat production potential of the Plains zebra by investigating carcass characteristics as well as meat and offal contributions. Dressing percentages were favourably comparable to equine, game, and livestock species. The considerable contribution of the edible by-products and the high muscle yield, indicate that the Plains zebra can potentially be used as a valuable protein source, thereby contributing to food security in especially rural areas of South Africa.

**Abstract:**

This study presents descriptive data on the meat and offal production of Plains zebras harvested in the winter (*n* = 8) and summer (*n* = 12) seasons, at different respective locations in the Western Cape Province of South Africa. The cold dressing percentages were numerically higher for the winter (58.0 ± 0.60%) than for the summer-harvested animals (56.60 ± 0.70%). Heavier internal offal yields were reported for the winter-harvested group (70.76 ± 21.8 kg) compared to the summer-harvested animals (66.13 ± 3.78 kg). As harvest season had no influence, pooled data of the percentage contribution of each muscle to cold carcass weight indicated that the *Longissimus et lumborum* (3.5 ± 0.18%), *semimembranosus* (1.6 ± 0.04%), *biceps femoris* (2.7 ± 0.05%) and *semitendinosus* (0.9 ± 0.02%) muscles differed significantly in their respective contributions to cold carcass weight. No significant differences were recorded between the contribution of the *infraspinatus* (0.6 ± 0.02%), *supraspinatus* (0.4 ± 0.03%) and *psoas major* (0.6 ± 0.02%) muscles. Carcass and muscle yields, as well as low-cost protein offal yields from this study indicate that the Plains zebra has the potential to be used and distributed as an alternative protein source.

## 1. Introduction

Meat derived from game animals is a well-known and upcoming protein source in South Africa. It forms part of the consumptive use in the game industry [1] and is actively being introduced in local well-known South African retail stores. Game species that are adapted to survive in harsh environments while producing quality meat, is considered suitable for meat production [2]. When farmed on marginal lands, game species produce meat with a higher return rate than conventional livestock farming [3,4]. The Plains zebra is physiologically and behaviourally adapted to survive in semi-arid conditions with low-quality forage [5]. They achieve this by having a hindgut digestive system that enables them to digest forage low in protein and high in structural carbohydrates, i.e., the long, tough stems and grass in the early stages of flush [5]. Plains zebra are capable of digesting cellulose at a faster rate than ruminants on the same terrain [6] and can extract more protein from poor quality grass than ruminants such as the wildebeest [5]. The Plains zebra is normally the first to venture into wetter, more wooded and taller pastures, acting as a pioneer for grassland biodiversity, and because of this, allows for the utilisation of forage by subsequent species such as antelope and wildebeest [5].

The Plains zebra can thus be considered as a good candidate for mixed species farming and can be maintained at higher stocking densities than similar-sized ruminants in grasslands of a poor nutritional quality [6]. A dry zebra mare of 4 years and 7 years is equal to 1.55 and 1.65 Large Animal Units (LAU), respectively, whereas stallions of 4 years and 7 years are equal to 1.40 and 1.45 LAU, respectively [7]. Based on these values, this amounts to an average of 1.52 zebras per LAU [7,8]. Bothma [8] calculated a more refined stocking density for zebras to be approximately 0.76 zebras per grazer unit or 0.63 zebras per browser unit. Due to the requirement to register properties that keep zebra in the African Horse Sickness Virus controlled area in South Africa (South African Animal Diseases Act, Act 35 of 1984), CapeNature was able to report 2498 Plains zebra in the Western Cape Province, spread across 212 holdings. It was thus calculated, out of 102 holdings in the Western Cape Province that Plains zebra is kept at a median density of 1.50 animals per km^2^ and a maximum of 71.4 animals per km^2^ [9].

The combined promotion of zebra meat and other familiar game species can potentially contribute to the promotion of game meat production, which will assist in the economic development of both the local and export game meat market supporting the drafted goals by the Game Meat Strategy of South Africa in 2022. The Plains zebra is not a cloven-hoofed animal and is not susceptible to the endemic foot-and-mouth disease (FMD), and therefore, they do not fall under the FMD control regulations. This makes them a suitable species for meat production and is now one of the leading export species from game abattoirs and meat exporters [10,11]. Zebras are susceptible to African Horse Sickness Virus, however, currently there is no regulation in place prohibiting meat exports. 

For game meat to compete with existing meat products, the meat production potential of various game species needs to be established to inform, not only the consumers, but meat processors and marketers as well. The first step to establishing the meat production potential of a game species is to obtain baseline information on slaughter traits such as carcass, muscle, and offal yields to determine the potential of such species for meat production purposes. Knowledge of these attributes in combination with nutritional and sensory attributes of the meat will help predict the value and the potential markets that the product can reach and be exported to. Information on the meat production of game species is limited, especially for Plains game such as the Plains zebra. 

According to Estes [5], the Plains zebra has the potential to be a commercially viable meat-producing animal due to its adaptation capability to various environments and large body size, with males weighing an average of 220–322 kg and females 175–250 kg. Research on carcass characteristics of the Plains zebra will enable marketers to be more informed about the potential of game meat as an alternative/substitute protein source, which will assist in marketing approaches that can aid in the development of new markets for game meat. The utilization of game as an additional protein source and improving the value chain can also potentially benefit food security in South Africa in an environmental and economically sustainable manner [3,10]. South Africa produces an insufficient amount of protein annually and needs to expand the utilisation of alternative local protein sources to meet the high demand because of the fast-growing population. Game animals are typically culled by professional teams with a rifle on game farms, with carcasses that are then transported to processing plants, from where the meat is distributed for commercial use [12]. South Africa has a strong hunting culture, which generates the potential for the utilisation of game meat, which in turn presents an economic incentive to the conservation of wildlife in South Africa [13]. Not only is the Plains zebra readily culled and hunted by professional teams and recreational hunters for their skins and meat, but it is also considered as a tourist attraction due to their unique phenotype. As a result, game animals such as the zebra, directly and indirectly, contribute to ecotourism, trophy hunting, breeding, game lodges, photographic safaris, and stock sales [3]. 

The abovementioned information indicates that a study on the meat production potential of Plains zebra, as quantified by carcass characteristics and meat quality parameters, is warranted to determine the potential contribution of this species to the economic viability and sustainability of the game industry in South Africa. Therefore, the aim of this study was to generate baseline information on carcass, muscle, and offal yields in the Plains zebra, which in turn may assist game farmers on the potential of this species to be used for meat production purposes. 

## 2. Materials and Methods

### 2.1. Animals and Study Location

#### 2.1.1. Winter Harvest: Prinskraal (June/Cold)

Eight Plains zebra stallions (adults of unknow age) were culled during June 2017 on Prinskraal farm (24°37′45.1″ S–20°06′44.9″ E) near Bredasdorp in the Western Cape Province, South Africa. The Plains zebras were mature and free-roaming and were cropped as part of the standard annual population control protocol of the farm. The farm is located in the Fynbos biome of South Africa, which is characterised by the presence of Central Rûens Shale Renosterveld that consist out of low to medium-tall grasses and dominated by renosterbos (*Elytropappus rhinocerotis*). Vegetation in this biome also includes *Aspalathus, Athanasia* and *Rhus* species [14]. The area receives an average annual precipitation of 300–480 mm, peaking from late autumn to the winter months (May to August) when 49% of the annual rains are received. The region is characterised by a maximum mean daily temperature of 27.3 °C in January, and a minimum of 5.6 °C in July [15]. 

Prior to the harvesting, the animals were maintained with 400 animals of various game species in a veld camp of approximately 800 ha in size. The zebras mostly foraged on Bermuda grass (*Cynodon dactylyn*) and the natural vegetation as described above. They also foraged on oat pastures when available. In the dry months, the zebras were given supplementary mineral licks. The farm was surrounded by exclusion fencing that prohibited animals from entering and exiting and there were no predators on the property. The animals were maintained for breeding purposes and for the landowner’s pleasure.

#### 2.1.2. Summer Harvest: Elandsberg Nature Reserve—Bartholomeus Klip (January/Warm-Hot)

Twelve male Plains zebra stallions of known age were culled during January 2018 in the Elandsberg Nature Reserve—Bartholomeus Klip (33°28′19.913″ S–19°2′18.916″ E), near Hermon in the Western Cape Province, South Africa. The Plains zebras for this harvest group consisted of sub-adult and adult Plains zebra stallions that were culled as part of the annual cropping program of the Quagga Project [16]. This study area is also located in the Fynbos biome but is dominated by the Swartland Alluvium Fynbos (SAF) veld type, ranging from the Elandskloof Mountains to the Limiet Mountains near Wellington. The SAF veldtype is known to be the wettest type of alluvium fynbos that is characterised by an average annual rainfall of between 320–980 mm. The area receives rain from May to August [15]. The region is also known to be the alluvium type of fynbos with the widest range of ambient temperatures, varying from an absolute maximum of 43 °C to an absolute minimum of 2 °C [17]. The veld is dominated by asteraceous fynbos, i.e., 0.7–0.9 m small-leaved proteoid and reed-like restioid species, and is characterized as being an evergreen fine-leaved shrub land with lengths varying from low to moderate [15,17]. 

The Plains zebras grazed on the natural vegetation until four to five months prior to harvesting. They were relocated from the veld camps to camps approximately 10 ha in size, and at a stocking density of four zebras per camp. There was no other animal sin the camps with the zebra. Available forage in the camps consisted primarily of Bermuda grass (*Cynodon dactylyn*), and to a lesser extent oat grass (*Avena sativa*) and ryegrass (*Lolium perenne*). The zebras were not given any supplementary feed or licks throughout the year. During the dry season (November to February) the grasses in the camps start to die off and becomes very fibrous with a low digestibility and protein content by January (when the Plains zebras were harvested). However, there were enough bulk left to sustain the zebras.

### 2.2. Plains Zebra Harvesting, Dressing and Sampling

Harvesting and evisceration procedures were conducted in accordance with the standard operating procedure (Ethical clearance number: 10NP_HOF02). Animals from both study locations were harvested during the day from a secure hunting vehicle by a sharpshooter with an appropriate rifle. The animals were culled with headshots to limit wounding and carcass damage/wastage [18]. Following the shot, all the animals were exsanguinated immediately and loaded onto the back of a designated vehicle. The time and date culled and additional notes regarding the shooting incident were documented to note pre-harvesting stress. The animals were transported to a nearby field slaughtering facility on the farm. At the slaughtering facility, the animals were loaded from the vehicle and weighed with a calibrated hanging scale to determine the exsanguinated, undressed weight of the animal. Due to the value of the skin, the skin was first removed, including that from the head before evisceration commenced.

The external (kin, hoofs, head), followed by the internal, offal were removed and weighed separately as described by van Schalkwyk & Hoffman [19]. The heart and kidneys were weighed after the removal of the surrounding fat layer. After evisceration, the carcasses were halved along the spinal column and divided between the second and last rib into four quarters separating the carcasses into two hindquarters (left and right) and two forequarters (left and right). The warm carcass quarters were weighed separately before entering a mobile chiller. The carcasses were placed in a suspended manner in the mobile chiller approximately 45 min post-mortem at ±4 °C and transported back to the Department of Animal Science at Stellenbosch University. Upon arrival, the carcasses were transferred to a chiller (±4 °C) located in the laboratory used for meat processing. 

Plains zebra harvested at Prinskraal had a ~72 h refrigeration period and Plains zebras harvested at Elandsberg Nature Reserved had a ~24 h refrigeration period before any carcass/muscle measurements were recorded. Following the refrigeration period, the cold carcass weights of each quarter and the ultimate pH were recorded. The carcasses were deboned, and seven commercially important muscles were excised from the left and right side of the carcasses. The weights of each muscle for the eight winter and 12 summer harvested Plains zebra stallions were recorded. The selected muscles included the fillets—*psoas major* (PM), two shoulder muscles—*infraspinatus* (IS) and *supraspinatus* (SS), three hindquarter muscles—*semimembranosus* (SM), *bicep femoris* (BF) and *semitendinosus* (ST), and lastly the *longissimus thoracis et lumborum* (LTL) which extends from the forequarter to the hindquarter along the vertebral column. The LTL were excised separately into the *longissimus thoracis* (LT) located in the forequarter and the *longissimus lumborum* (LL) located in the hindquarter. 

### 2.3. Statistical Analysis

The statistical analysis was conducted using Statistica 64 version 13.4 (2018) VEPAC model and Microsoft Excel (2016). Descriptive statistics were used as an indication of the expected variation for the carcass yields, offal yields and muscle yields per season. An exponential regression analysis was performed to determine the relationship between the offal yields and age for Plains zebras harvested in the summer season as well as between the proportional contribution of each muscle and cold carcass weight from both seasons. The contribution for each selected muscle to the cold carcass weight from both seasons was pooled and analysed by means of univariate analysis of variance (ANOVA), with muscles as a fixed effect and animals as a random effect. A 5% significance level was used.

## 3. Results

Only observations on the carcass, offal, and muscle yields between the two seasons could be made due to the descriptive statistical method used to quantify the data. The ages of the winter group were unknown whilst that of the 12 stallions from the summer group were known as they were cull stallions from the Quagga breeding project; the latter allows for a comparison of the effect of age (even though the numbers are low) on the yields. Additionally, the Plains zebra harvested in the winter season were classified as physically mature adults over the age of three years [20]. The known ages for animals harvested in the summer season ranged from two to 13 years.

### 3.1. Carcass Yields

Table 1 presents the carcass parameters of Plains zebra stallions harvested during two seasons in the Western Cape Province of South Africa. The undressed weight of animals harvested in the winter had a mean of 324.4 ± 5.55 kg with a minimum of 304.8 kg and a maximum of 353.7 kg. Animals in the summer season were found to have a numerically lower mean undressed weight of 291.5 ± 11.65 kg with a minimum of 234.4 kg and a maximum of 347.8 kg. The winter harvesting group had a mean warm carcass weight of 193.1 ± 3.95 kg and a cold carcass weight of 188.3 ± 4.03 kg. As expected, the warm (168.9 ± 5.79 kg) and cold carcass weights (164.5 ± 5.53 kg) for the summer harvesting group were also found to be numerically lower than the winter harvesting group due to the lower undressed weight found for the summer group. The winter harvesting group had a warm dressing percentage of 59.5 ± 0.55% and a cold dressing percentage of 58.0 ± 0.60%. The summer harvesting group had a warm dressing percentage of 58.1 ± 0.68% and a cold dressing percentage of 56.6 ± 0.70%. Both harvesting groups had a moisture loss between the warm and cold dressing percentage of ~1.5%. 

A regression was calculated to determine the exponential relationship between the carcass characteristics and age of the summer harvesting group. The coefficient of determination (R^2^ value) and the exponential equation for all the carcass parameters are presented in Table 2. The undressed, warm, and cold carcass weights had weak R^2^ values of 0.2311, 0.2369 and 0.2460, respectively. The warm and cold dressing percentage also had a weak R^2^ values of 0.0378 and 0.0583, respectively. This indicates that the correlation between the carcass yields, and age is low.

### 3.2. Offal Yields

The mean external and internal offal weights (kg) and its contribution to the undressed weight (%) for Plains zebra stallions harvested in the winter and summer season are presented in Table 3. Included in the external offal is the head, legs, skin, and tail and was found to be numerically similar for the winter (41.980 ± 0.853 kg) and summer harvest (41.950 ± 1.334 kg) which contributed 13.0 ± 0.23% and 14.5 ± 0.29% to the undressed weight, respectively. The filled gastrointestinal tract (GIT), liver, heart, trachea, lungs, kidneys, and spleen made up the internal offal component of the carcass. The internal offal of Plains zebras harvested in the winter season weighed 70.760 ± 3.113 kg and contributed 21.8 ± 0.89% to the undressed weight. The summer harvest group had a numerically lower internal offal weight of 66.132 ± 3.782 kg with an almost similar contribution of 22.5 ± 0.50% to the undressed weight. The total skin weight and contribution thereof to the undressed weight of the Plains zebra harvested in the winter season (22.288 ± 0.533 kg; 6.9 ± 0.19%) were found to be lower than for the Plains zebras harvested in the summer season (24.046 ± 0.927 kg; 8.3 ± 0.18%). The liver weight was found to be numerically similar for the Plains zebras harvested in the winter (3.300 ± 0.130 kg) and summer season (3.320 ± 0.109 kg) which contributed 1.0 ± 0.04% and 1.2 ± 0.05% to the undressed weight, respectively. The kidneys had a numerically similar mean contribution for both seasons (0.2 ± 0.01%) and weighed 0.740 ± 0.030 kg and 0.648 ± 0.032 kg for the winter and summer harvest, respectively. The total offal for the winter harvesting group was 112.740 ± 3.424 kg and for the summer was 108.081 ± 4.933 kg. The total offal comprised 34.8 ± 0.94% and 37.1 ± 0.44% of the undressed weight for the winter and summer harvesting group, respectively. 

Table 4 represents the coefficient of determination (R^2^ value) and the equation for an exponential regression drawn for the proportion of the offal (%) over the age of the 12 Plains zebras in the summer harvesting group. Only 6.50% (R^2^ = 0.0650) of the variation in the total offal can be attributed to the exponential relationship with age.

### 3.3. Muscle Yields

The mean muscle weights (kg) and their contribution to the cold carcass weight (%) for Plains zebra stallions harvested in the winter and summer season are presented in Table 5. The mean weight is the combination of the seven muscles on the left and right side of the carcass. The muscle weights of the winter harvest group were numerically higher than the summer harvest group. The LTL muscle was found to have the highest (*p* < 0.001) contribution of 3.9 ± 0.39% and 3.3 ± 0.30% to the cold carcass weight for the winter and summer season, respectively. The SS muscle was found to have the lowest contribution and contributed numerically similar to the cold carcass weight for both the winter (0.4 ± 0.06%) and summer (0.4 ± 0.01%) seasons. The total mean contribution to cold carcass weight of all seven muscles was 11.0 ± 0.43% and 9.9 ± 0.16% for the winter and summer groups, respectively.

A regression was calculated to determine the relationship between the muscle yields (kg and %) and the cold carcass weights for both harvesting groups. The coefficient of determination (R^2^-value) and the exponential equation for all the muscle yields are presented in Table 6. The variation of the total muscle weight relative to the cold carcass weight taken into consideration by the equation was 78.61% (R^2^ = 0.7861).

Figure 1 represents the mean proportion (%) of each individual muscle to the mean cold carcass weight (176.4 kg) of 20 Plains zebra stallions. The LTL (3.5%), SM (1.6%), BF (2.7%) and ST (0.9%) muscles’ contributions differed significantly from one another. The LTL muscle had the highest contribution followed by the BF and then SM. The proportion of the IS (0.6%), SS (0.4%) and PM (0.6%) to the cold carcass weight were non-significant.

## 4. Discussion

### 4.1. Carcass Yields

It was observed that the animals harvested in the summer season (291.5 ± 11.65 kg) had a lower undressed weight than the animals harvested in the winter season (324.4 ± 5.55 kg; Table 1). The differences in undressed carcass weight were reflected in the lower dressed carcass yields (warm and cold) observed for the summer-harvested animals (warm = 168.9 ± 5.79 kg; cold = 164.5 ± 5.53 kg) compared to the winter-harvested animals (warm = 193.1 ± 3.95 kg; cold = 188.3 ± 4.03 kg). Information on the carcass weight of the Plains zebra is limited to two studies [21,22]. Both these studies reported lower carcass yields compared to the current study, which may potentially be attributed to the dry season the animals in these two studies were harvested in. The lower carcass yields reported for the summer-harvested animals in the current study could be due to seasonal differences in the diet, harvest location and forage behaviour, altering the plain of nutrition and as a result the body composition. In the Western Cape Province, the rainy season is in the winter between May and August and the dry summer months between November and February. During the wet season, the availability of forage increases in biomass and consists of leafy grass with high nutrient concentrations. In contrast, the biomass during the dry season is reduced and the Plains zebras are dependent on fibrous grass species with low nutritional quality [23]. Additionally, the Plains zebras harvested in the winter season, in this study, had access to a supplement lick during the dry months to restore depleted minerals. Furthermore, the Plains zebra also tends to consume more forage in the rainy season, than in the dry season, due to natural grazing being more abundant, leading to more grass bites per step taken [24]. This results in an increase of fat deposition in the rainy season and as observed in this study, the undressed and dressed carcass weights were heavier for the Plains zebras harvested in the winter season. Visually, it was noted in this study that the winter-harvested Plains zebra carcasses were characterised by a higher degree of subcutaneous fat deposition, when compared to that of the summer-harvested Plains zebras. However, it needs to be kept in mind that the difference in level of subcutaneous fat deposition may potentially also be ascribed to the mixed age composition of the summer-harvested group (i.e., sub-adults and adults), whereas the winter-harvested group consisted of only adult animals 

Historically, meat-producing horses were slaughtered at the end of their working career, and only recently this practice was changed with foals being slaughtered for consumption [25]. The slaughter of meat-producing horses is thus based on age which have a beneficial influence on carcass characteristics, with season not considered as these horses are provided with artificial feed year-round irrespective of the location or season [26]. In this study, to investigate the effect of age on undressed, warm and cold carcass weights, and dressing percentage for Plains zebra, an exponential regression analysis was carried out. A low coefficient of determination (R^2^ ≤ 0.24) was reported for all the carcass factors, indicating that less than 24% of the variation in carcass yields can be ascribed to the influence of animal age. The low R^2^ value potentially indicate that the summer-harvested animals already achieved maximum growth and thus were in the plateau phase on a typical sigmoidal curve at the point of harvest. Mentis [20] found that free-roaming Plains zebra reach their mature body weight at the age of 36 months. The youngest animal in this study was 24 months, and the advanced growth may potentially be ascribed to the fact that these animals were part of the Quagga project, i.e., would have received additional feed when required and thus not have been subjected to the challenges of seasonal forage availability and quality that free-roaming Plains zebras experience. Thus, the differences found between the two groups in this study are most likely due to the effect of the harvesting season on subcutaneous fat deposition and, as a result, body composition.

Little information is available on the meat production potential of Plains zebra, therefore findings in this study will be compared to related information about commercial horse meat breeds and/or donkey breeds. With a focus on specialized horse breeds for meat production, most of the breeds had higher slaughter and carcass weights than the Plains zebra. Heavier weights were reported for the Burguete horse breed slaughtered at 16 months (411.3 kg and 275.5 kg, respectively) [27] and 24 months (395 kg and 258.9 kg, respectively), the Hispano-Bretón (HB) breed at 24 months (406 kg and 275.5 kg, respectively) [28], and Sanfratellano breed at 18 months (411 kg and 243.75 kg) [29]. The Haflinger breed also slaughtered at 18 months had a comparable slaughter weight (349.83 kg) and carcass weight (207.83 kg) to the Plains zebras in this study [29]. In this Plains zebra study, it was noted that there is little correlation between the carcass yields and age; similarly, no significant differences were found between age and carcass weights in slaughter horses between the ages of three and 21 years. The average warm carcass weights of these slaughtered horses were 586.4 kg for geldings, 663.5 kg for mares and 671.5 kg for stallions [30]. The Plains zebra is heavier than Martina Franca donkeys (101–181 kg; 8–15 months of age) [31,32,33] and domesticated donkeys found in tropical regions in Africa (such as Botswana), Central America and Asia (<150 kg) [34]. Interestingly the undressed and cold carcass weight of the Plains zebra harvested in the summer season are comparable to Martina Franca × Ragusana male donkeys (285 kg and 154 kg, respectively) whilst those harvested in the winter season to male mules (363 kg and 212 kg, respectively) slaughtered at 16 years of age [35]. Compared to the meat-producing horse and donkey breeds, the Plains zebras in this study are intermediate in terms of undressed-, warm- and cold carcass weights. 

Dressing percentage is an important parameter when the meat production potential of an animal or species is investigated [36]. In game animals, the dressing percentage is the proportion of the dressed weight relative to the undressed carcass weight. Game animals cannot be fasted before harvesting and therefore may have a higher gut fill than domestic livestock species that are typically fasted in lairage for 24 h before slaughter. Comparable results for the dressing percentage for both the winter (59.5 ± 0.55%) and summer seasons (58.1 ± 0.68%) were found in this study. The difference observed is attributed to the higher carcass yields observed and the higher subcutaneous fat level noted in the winter harvesting group. A lower dressing percentage of 56% for Plains zebra was reported [21], however, the sample size of the study was very small (*n* = 2) and animals were harvested in very dry, semi-arid conditions; these conditions may have contributed to the low carcass yields and dressing percentages reported. 

Meat-producing horses typically yield high dressing percentages, when compared to other domestic livestock species. The dressing percentage of the Plains zebras in this study were higher than that reported for the HB × Galician Mountain crossbreed (52.8%) and the Galician Mountain breed (50.3%) [37], and are comparable to that of the Haflinger (59.6%) and Sanfratellano horse breeds (59.3%) [29]. However, horse breeds such as the Burguete [27,28], HB [28], Italian Heavy Draft horse breed [38] and horses slaughtered in Poland [39] had higher dressing percentages (>63%) than the Plains zebras harvested in this study. The studies reporting high dressing percentages can potentially be ascribed to a low gut fill as a result of the animals subjected to pre-slaughter fasting, which subsequently resulted in higher dressing percentages. Another factor that can possibly influenced the dressing percentage reported for Plains zebra in this study, is the degree of blood loss before weighing; potentially resulting in a higher calculated dressing percentage. The dressing percentage of the Plains zebras harvested in both seasons compared favourably to that of Martina Franca donkeys (49.2–57.5%) found in various studies [32,33,34]. The warm and cold dressing percentage of the Plains zebra also compared favourably to Martina Franca × Ragusana male donkey foals reared intensively (55.2% and 54.3%, respectively) and extensively (53.1% and 51.3%, respectively) [40].

When comparing the dressing yield of the Plains zebra with other game species commonly found in South Africa, the dressing percentage compared favourably with impala (58.0%) [41], springbok (56%) [42], blesbok (50.6–53.7%) [42,43] and large-bodied game species such as the greater kudu (58.3%) [44], gemsbok (54%) [21], eland (50.8–51%) [42,45], black wildebeest (53.19%) [46], African savannah buffalo (48–53%) [47] and male giraffe (51.6–59.2%) [48]. However, higher dressing percentages for male springbok (64.9%), blesbok (62.9%) and impala (63.4%) have been reported [49]. The differences in dressing percentages can be attributed to seasonal and location differences influencing the body composition of these species [50]. 

The dressing percentages of game animals are also comparable with domestic livestock species such as cattle, sheep, and goats. The dressing percentage of the Plains zebra compares favourably to that of Nguni (52.1%), Bonsmara (56.9%) and Aberdeen Angus (53.7%) cattle [51]. The Plains zebra values in this regard also compared favourably to sheep breeds such as the South African Mutton Merino (SAMM; 41.5%), Dormer (44.2%) [52], and fat-tailed indigenous Damara (59.9%) and also to goats such as Boer goats and indigenous goats (55.7%) [53].

### 4.2. Offal Yields

Game animals produce by-products that are edible and non-edible. Edible by-products include tripe, liver, heart, lungs, kidneys and spleen; and external offal which include the head, tail and feet/lower legs [54]. Edible offal is marketed by informal suppliers and forms part of the traditional diet in South Africa [54] with a per capita consumption of 4.7 kg and 4.8 kg for the year of 2009 and 2013, respectively [1]. Information on the quantity of edible offal of game species is limited and needs to be investigated since edible offal is a low-cost protein and nutrient-rich food source that has the potential to contribute to food security [54,55]. Non-edible offal, in game species, such as the skin and horns (as well as the head after removal of the cheek and other meat/muscles) are primary products collected during trophy hunting and are of high economic value in the game industry [54,56]. The Plains zebra is a hindgut fermenter and lacks horns, unlike some game species, and should therefore rather be compared with other equine species in terms of head and GIT weights. Information on offal yields in horses and donkeys, however, is limited. 

In this study, the highest contributing external offal component to the undressed carcass weight in Plains zebras harvested in both locations, was the skin followed by the head (Table 3). Presently, low quantities of Plains zebra meat enter the formal market value chain in South Africa due to Plains zebras being primarily hunted by recreational hunters for their skins with the meat seen as a secondary product. However, large numbers of zebras are culled by professional teams for the utilisation of their skin and meat. The meat is typically exported to the European Union as Plains zebras are excluded from the FMD control regulation [57]. It is important for the industry to note the contribution of the skin and head (as well as the legs under the knee joint with the hooves) to the carcass weight as these body parts remain attached to the carcass during transportation to the processing plants. This enables the marketers to achieve a top price for the skin since the skin on the head is also flayed with that of the body and lower legs. Another reason for keeping the skin on the carcass during transportation and chilling, is to reduce moisture loss since only a few game species have a subcutaneous fat layer to minimise this loss. However, in this study, the head and skin were removed prior to transportation and chilling and the head of the Plains zebras from both locations was weighed without the skin as the skin was flayed according to industry standards. A lower skin weight and contribution to the undressed weight for animals harvested in the winter season (22.288 ± 0.533 kg; 6.9 ± 0.19%) were found than for animals harvested in the summer season (24.046 ± 0.927 kg; 8.3 ± 0.18%). The lower skin weight of animals harvested in the winter than in the summer can potentially be attributed to different skinning teams that were used for each season. A professional skinning team were used for the winter harvest, and a non-professional skinning team were used for the summer harvest, which may have resulted in variations in the subcutaneous fat left on the skin, thus influencing the contribution of the skin to the undressed weight. 

The head weight and its contribution to the undressed weight were found to be very similar for the winter (12.606 ± 0.322 kg; 3.9 ± 0.05%) and summer-harvested animals (12.046 ± 0.344 kg; 4.2 ± 0.13%). A weak relationship between animal age and head and skin weight, as evident in the low R^2^ value obtained with the exponential regression analysis, is indicative that the animals were mature at the point of harvest, which supports the conclusion for the carcass yields. These findings agree with the observations in male horses slaughtered in México where the skin contributed 7.4% and the head 4.5% to the live weight (277.8 kg) [58]. The slaughter weight of these horses fell in the range of the undressed weight of the Plains zebras harvested in the summer season (Table 1). Catalan crossbreed donkeys with a live weight of more than 151 kg were found to have a similar head and lower skin contribution to the empty body weight (live weight−gastrointestinal content) than the Plains zebras in the current study [59]. The lower skin contribution observed can be postulated to be due to the skin weight not including the skin on the head and legs, as the study did not provide a detailed methodology [59]. 

When compared to smaller game species, the proportional weight of the skin of the Plains zebra from both seasons was higher than found for male impala (3.9%) and male fallow deer (6.6%) with an undressed weight of 49.9 kg and 47.4 kg, respectively [41,60]. The skin weight and contribution can also be compared to large-bodied game species which will have heavier skins and body weights such as eland bulls (14.3 kg and 6.8%) with an undressed weight of 208.6 kg [61] and blue wildebeest bulls (13.6–17.9 kg and 7.9–8.4%) weighing 168.8–208.2 kg [62]. 

In terms of the internal offal, animals harvested in the winter season (70.760 ± 3.113 kg; 21.8 ± 0.89%) produced heavier or similar yields when compared to animals harvested in the summer season (66.132 ± 3.782 kg; 22.5 ± 0.50%). The GIT had the highest internal weight and contribution to the undressed weights of animals harvested in the winter (60.150 ± 3.300 kg; 18.6 ± 0.97%) and summer season (56.050 ± 3.547 kg; 19.1 ± 0.54%). The lower GIT weight reported for the summer-harvested animals can potentially be attributed to a lower gut fill, since it has been shown by the exponential regression that the effect of age had a weak effect on the GIT proportion (R^2^-value = 0.1153; Table 4). It has been found that in the rainy (winter) season the Plains zebra consumes more forage (by counting the grass bites per step taken) because of the natural grazing being more abundant [24]. 

The liver represents a valuable source of vitamin A, vitamin B_1,_ and nicotinic acid. The liver and kidneys contain higher levels of iron, copper and zinc than skeletal muscle, [63], making it a valuable low-cost protein source especially for the rural communities of South Africa [55]. However, recently offal is rather processed into niche-market products that are seen as delicacies in upmarket restaurants [55]. The liver weight and contribution to total carcass yield in the winter season (3.300 ± 0.130 kg; 1.0 ± 0.04%) were comparable with that found for the summer season (3.320 ± 0.109 kg; 1.2 ± 0.5%). The kidney weights in the winter season (0.740 ± 0.030 kg; 0.3 ±0.01% of carcass weight) were also comparable with that found for the summer season (0.648 ± 0.032 kg; 0.2 ± 0.03%). The contribution of the liver weight to the empty body weight (live weight−gastrointestinal content) in Catalan crossbreed donkeys with a live weight of >151 kg was found to be similar to the Plains zebras when calculated in the same manner [59]. 

The liver weight as a proportion to carcass weight is comparable to that of impala (1.3%; [41]), blesbok (1.3%; [49]) and blue wildebeest (1.0%; [62]). When compared to livestock species such as the South Africa Mutton Merino (SAMM; 0.99 kg) and Dormer lambs (1.13 kg) [52], the Plains zebra has heavier livers. However, the contribution to the undressed weight was similar to SAMM (1.9%) and Dormer (2.0%) sheep breeds [52]. The proportional weight of the kidneys relative to the live weight are comparable with impala (0.3%; [41]), fallow deer (0.3%; [60]), blesbok (0.3%; [49]) and blue wildebeest (0.2%; [62]). 

### 4.3. Muscle Yields

The same primal meat cuts used to section beef carcasses are also used for the sectioning of game meat species and meat-horse breeds. The game meat trade however tends to sell muscles whole rather than as meat cuts that are frequently made up out of two or more muscles. In this study, seven muscles including the LTL (loin), SM (topside), BF (silverside), ST (eye of the round), PM (fillet) and two additional representative shoulder muscles, the SS and IS, were removed from the carcass and weighed to determine each muscle’s contribution to the overall meat yield in Plains zebra. To the best of our knowledge, no information is available on the contribution of the respective muscles to the meat yield in Plains zebra. 

Recorded weight for all seven muscles for the winter-harvested animals were numerically heavier when compared to the weights recorded for the summer-harvested group (Table 5). Notable numerical differences between the winter and summer harvest can be observed for the LTL (7.455 ± 0.829 kg and 5.410 ± 0.298 kg, respectively), SM (3.178 ± 0.093 kg and 2.642 ± 0.120 kg, respectively) and BF (5.271 ± 0.188 kg and 4.215 ± 0.142 kg, respectively) muscles. The difference in weight between the two seasons can potentially be attributed to the lower carcass yields observed for the summer-harvested group (Table 1), due to the real muscle yield being dependent on the weight at slaughter and consequently the carcass weight. The remainder of the seven muscles had almost numerically similar weights and proportions when compared between seasons. Age did not have a strong influence on carcass or organ yields observed for the summer-harvested animals, and therefore an exponential regression analysis was carried for each muscle weight and its calculated proportion to the cold carcass weight. The analysis generated high R^2^ values for the LTL (0.6501), BF (0.7466) and ST (0.6431), however, it is more of value to consider the proportional contribution of each muscle to the cold carcass weight. The R^2^-values showed a weak increase with the cold carcass weight with 29.1% (highest R^2^-value) of the variation in the LTL and 0.4% (lowest R^2^-value) of the variation in the ST explained by the exponential relationship with the cold carcass weight. The results indicates that the proportional contribution of the muscles is relatively similar irrespective of the lower or higher carcass weights observed in this study. 

The mean proportion (%) of each individual muscle to the mean cold carcass weight of the 20 Plains zebra stallions were pooled and presented in Figure 1. Significant differences were found between the hindquarter muscles; the LTL (3.5%), SM (1.6%), BF (2.7%) and ST (0.9%) which also significantly differed from the smaller sized muscles, the IS (0.6%), SS (0.4%) and PM (0.6%). The differences found between muscles in terms of contribution (and weight) can be attributed to their anatomical location and function. The LTL was found to be the highest contributing muscle which is expected since the LTL is an epaxial muscle with many bundle fibres extending over the vertebral column from the forequarter to the hindquarter [64]. The BF was the second-highest contributing muscle followed by the SM and then the ST. These muscles are the proximal muscles on the pelvic limb and are larger than the muscles on the shoulder joint due to their locomotive and hip extension functions [64,65]. The PM muscle is a small muscle involved in the flexion of the hip and is one of the most tender muscles on a carcass due to its low functional activity resulting in small fibre diameters and less connective tissue. The PM muscle was found to have a similar contribution as the two stabilizing shoulder muscles; the IS and the SS which are respectively involved in the flexion and extension of the shoulder [64]. 

The muscle yields of the Plains zebra can be compared to game species and various horse breeds. However, data on individual muscle yields in various horse breeds is limited, since yields are represented as primal meat cuts consisting out of two or more muscles and therefore cannot be compared. As most meat-producing horse breeds are larger at slaughter, one can speculate that muscles represented in primal meat cuts in the Plains zebra will be smaller than the corresponding muscles in horses, and bigger than the corresponding muscles in donkeys. The LTL, SM, BF and ST in the Plains zebra were found to have a lower contribution to the total carcass weight, when compared to findings reported for blue wildebeest (4.8–5.0%, 3.8–3.9%, 4.8–5.0% and 1.5%, respectively) [62] and fallow deer (7.5%, 5.2%, 6.1% and 1.6%, respectively) [60]. The LTL muscle had a higher contribution, the SM a lower contribution, the BF a higher contribution and the ST were found to have a similar contribution to the cold carcass weight when compared to eland (2.6%, 2.2%, 2.4% and 0.9%) [45]. Higher percentages for the IS and SS muscle were reported for fallow deer (1.3% and 1.1%, respectively) [60], eland (0.7% and 0.7%, respectively) [45] and blue wildebeest (1.3–1.4% and 1.1–1.2%, respectively) [62]; results that may be related to a more active behaviour, e.g., like jumping more frequently by fallow deer and eland. 

## 5. Conclusions

Due to the limited zebra numbers and the fact that of some, the ages were unknown, may have resulted in the data being confounded and thus the findings need to be treated with caution. None the less, the study is the first of its kind to provide baseline information on the carcass, offal and muscle yield potential of Plains zebras harvested in two localities in the Western Cape Province of South Africa. The study included animals that were harvested during two seasons to establish a potential influence of season on the carcass characteristics of Plains zebra. Carcass dressing percentage compared favourably and in agreement with other equine species, game species and livestock species such as cattle and sheep. The considerable contribution of the edible by-products in this study and the high muscle yield, indicate that the Plains zebra can potentially be used as a valuable animal protein source, thereby contributing to food security in especially rural areas of South Africa.

Further studies are warranted to determine the full production potential and economic viability of the Plains zebra. Future studies can investigate forequarter and hindquarter commercial meat cuts in terms of weight and percentage to the cold carcass weight for a better comparison to respectively horse, donkey, and cattle commercial meat cuts. Studies on the meat and bone yield and meat-to-bone ratios and percentages are also warranted, for there is no information available in literature on these aspects. Other factors that also need to be investigated include, the influence of production systems, season, ages and sex on the meat production potential of Plains zebra. As Plains zebra meat is exported regularly to the EU it will be of benefit to establish the physical meat quality as well as nutritional and organoleptic attributes of the meat to promote further marketing as an alternative/substitute meat product. 

## Figures and Tables

**Figure 1 animals-13-00050-f001:**
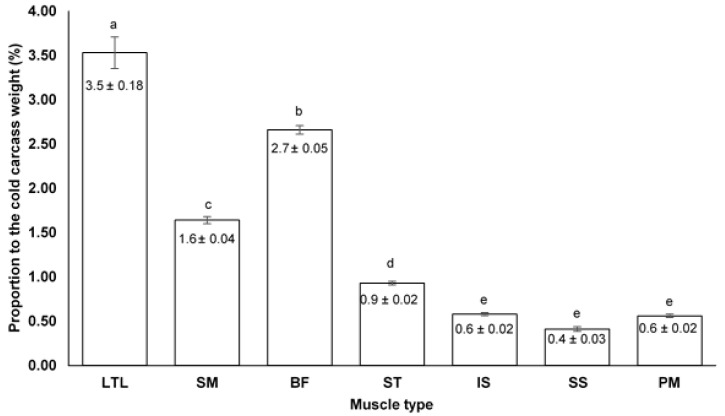
The percentage contribution (mean ± SE) of the *longissimus thoracis et lumborum* (LTL), *semimembranosus* (SM), *biceps femoris* (BF), *semitendinosus* (ST), *infraspinatus* (IS), *supraspinatus* (SS) and *psoas major* (PM) muscles to the average cold carcass weight of Plains zebra stallions harvested during two seasons in the Western Cape Province of South Africa. ^a–e^ Means with different letters differ (*p* < 0.05).

**Table 1 animals-13-00050-t001:** Mean (± standard error) and the range of carcass yields from eight Plains zebra stallions harvested in the winter season and 12 Plains zebra stallions harvested in the summer season.

Carcass Parameter		Winter Harvest (*n* = 8)		Summer Harvest (*n* = 12)	
		Mean ± SE	Range	Mean ± SE	Range
Undressed carcass weight	kg	324.4 ± 5.55	304.8–353.7	291.5 ± 11.65	234.4–347.8
Warm carcass weight	kg	193.1 ± 3.95	176.0–210.1	168.9 ± 5.79	139.3–190.8
Cold carcass weight	kg	188.3 ± 4.03	169.6–205.0	164.5 ± 5.53	136.1–185.5
Warm dressing percentage	%	59.5 ± 0.55	57.7–62.0	58.1 ± 0.68	54.9–63.2
Cold dressing percentage	%	58.0 ± 0.60	55.6–60.7	56.6 ± 0.70	53.3–61.2

Abbreviations: SE= standard error.

**Table 2 animals-13-00050-t002:** The coefficient of determination (R^2^-value) and the exponential equation to indicate the relationship between the respective carcass parameters and age of Plains zebra stallions (aged between 2 and 13 years) harvested during a summer season in the Western Cape Province of South Africa.

Carcass Component	Exponential Equation	R^2^-Value
Undressed weight	y = 262.39e^0.0167x^	0.2311
Cold carcass weight	y = 150.58e^0.0142x^	0.2369
Warm carcass weight	y = 154.05e^0.0148x^	0.2460
Warm dressing percentage	y = 58.709e^−0.002x^	0.0378
Cold dressing percentage	y = 57.386$e^{-2\times 10^{-4}x}$	0.0583

y = carcass component; x = age.

**Table 3 animals-13-00050-t003:** Mean (±standard error), minimum and maximum of offal yields from eight Plains zebra stallions harvested in the winter season and 12 Plains zebra stallions harvested in the summer season.

Carcass Components	Winter Harvest (*n* = 8)			Summer Harvest (*n* = 12)		
	Mean ± SE	Mean ± SE	Range	Mean ± SE	Mean ± SE	Range
	(kg)	(% ^1^)	(kg)	(kg)	(% ^1^)	(kg)
Undressed weight	324.4 ± 5.55		304.8–353.7	291.5 ± 11.65		234.4–347.8
Head ^2^	12.606 ± 0.322	3.9 ± 0.05	11.70–14.45	12.046 ± 0.344	4.17 ± 0.13	10.30–14.06
Legs	6.725 ± 0.182	2.1 ± 0.04	6.000–7.550	5.408 ± 0.159	1.867 ± 0.04	4.500–6.100
Skin	22.288 ± 0.533	6.9 ± 0.19	20.800–25.100	24.046 ± 0.927	8.3 ± 0.18	18.350–29.200
Tail ^3^	0.362 ± 0.040	0.1 ± 0.01	0.241–0.600	0.449 ± 0.029	0.2 ± 0.01	0.314–0.645
Total external offal	41.98 ± 0.853	13.0 ± 0.23	39.39–46.39	41.95 ± 1.334	14.5 ± 0.29	34.08–49.93
GIT	60.150 ± 3.300	18.6 ± 0.97	48.000–76.200	56.050 ± 3.547	19.1 ± 0.54	37.450–74.850
Liver	3.300 ± 0.130	1.0 ± 0.04	2.750–3.750	3.320 ± 0.109	1.2 ± 0.05	2.600–3.900
Heart	1.788 ± 0.093	0.6 ± 0.03	1.350–2.100	1.727 ± 0.084	0.6 ± 0.02	1.309–2.266
Trachea & Lungs	3.850 ± 0.458	1.2 ± 0.13	2.600–6.000	3.417 ± 0.187	1.2 ± 0.06	2.500–4.550
Kidneys	0.740 ± 0.030	0.2 ± 0.01	0.661–0.904	0.648 ± 0.032	0.2 ± 0.01	0.487–0.860
Spleen	0.933 ± 0.043	0.3 ± 0.01	0.776–1.113	0.969 ± 0.074	0.3 ± 0.07	0.631–1.586
Total internal offal	70.760 ± 3.113	21.8 ± 0.89	59.393–86.558	66.132 ± 3.782	22.5 ± 0.50	46.453–86.360
Total offal	112.74 ± 3.424	34.8 ± 0.94	100.44–131.02	108.08 ± 4.933	37.1 ± 0.44	80.63–136.29

^1^ Parameter as a percentage of the undressed weight, ^2^ Head: Measured without skin and with tongue, ^3^ Tail: Measured without skin. Abbreviations: SE= standard error, GIT = Gastrointestinal tract.

**Table 4 animals-13-00050-t004:** The coefficient of determination (R^2^ value) and the exponential equation to indicate the relationship between the respective offal components and age of Plains zebra stallions (aged between 2 and 13 years) harvested during a summer season in the Western Cape Province of South Africa.

Offal Components	Exponential Equation	R^2^ Value
Head	y = 4.2378$e^{-3\times 10^{-4}x}$	0.0180
Legs	y = 1.8479e^0.0001x^	0.0060
Skin	y = 8.4970$e^{-4\times 10^{-4}x}$	0.0767
Tail	y = 0.1588$e^{-6\times 10^{-4}x}$	0.0244
Total external offal	y = 14.754$e^{-3\times 10^{-4}x}$	0.0494
GIT	y = 18.078e^0.0007x^	0.1153
Liver	y = 1.2215$e^{-1\times 10^{-3}x}$	0.1092
Heart	y = 0.5669e^0.0006x^	0.0784
Trachea & Lungs	y = 1.0555e^0.0014x^	0.1731
Kidneys	y = 0.2369$e^{-1\times 10^{-3}x}$	0.2238
Spleen	y = 0.3643e^−0.002x^	0.1472
Total internal offal	y = 21.594e^0.0006x^	0.1350
Total offal	y = 36.419e^0.0002x^	0.0650

**Table 5 animals-13-00050-t005:** Muscle yields from eight Plains zebra stallions harvested in the winter and 12 Plains zebra stallions harvested in the summer season. The results are reported as Means (± standard error), minimum and maximum of the muscle weight (kg) and contribution (%) to the cold carcass weight.

Carcass Components	Winter Harvest (*n* = 8)			Summer Harvest (*n* = 12)		
	Mean ± SE	Mean ± SE	Range	Mean ± SE	Mean ± SE	Range
	(kg)	(% ^1^)	(kg)	(kg)	(% ^1^)	(kg)
CCW	188.3 ± 4.03	-	169.6–205.0	164.5 ± 5.53	-	136.1–185.5
LTL	7.455 ± 0.829	3.9 ± 0.39	4.675–11.025	5.410 ± 0.298	3.3 ± 0.10	3.825–7.275
SM	3.178 ± 0.093	1.7 ± 0.03	2.875–3.600	2.642 ± 0.120	1.6 ± 0.06	1.800–3.275
BF	5.271 ± 0.188	2.8 ± 0.07	4.625–6.075	4.215 ± 0.142	2.6 ±0.05	3.475–4.928
ST	1.819 ± 0.106	1.0 ± 0.04	1.425–2.200	1.490 ± 0.049	0.9 ± 0.02	1.300–1.725
IS	1.069 ± 0.053	0.6 ± 0.03	0.925–1.350	0.971 ± 0.042	0.6 ± 0.02	0.775–1.200
SS	0.841 ± 0.122	0.4 ± 0.06	0.600–1.675	0.633 ± 0.031	0.4 ± 0.01	0.475–0.800
PM	1.084 ± 0.039	0.6 ± 0.02	0.900–1.200	0.902 ± 0.044	0.6 ± 0.03	0.600–1.250
Total	20.72 ± 1.117	11.0 ± 0.43	16.77–24.15	16.26 ± 0.581	9.9 ± 0.16	13.53–19.68

^1^ Parameter as a percentage to the cold carcass weight. Abbreviations: CCW = Cold Carcass Weight, LTL= longissimus thoracis et lumborum, BF= biceps femoris, SM = semimembranosus, ST= semitendinosus, IS = infraspinatus, SS = supraspinatus, PM= psoas major, SE= standard error.

**Table 6 animals-13-00050-t006:** The coefficient of determination (R^2^ value) and the exponential equation of the muscle yields drawn over the cold carcass weights for 20 Plains zebras harvested in the winter and summer season.

Carcass Components	Weight		Proportional Contribution	
	Exponential Equation	R^2^ Value	Exponential Equation	R^2^ Value
LTL	y = 0.8205e^0.0114x^	0.6501	y = 1.3448e^0.0054x^	0.2914
SM	y = 1.0904e^0.0055x^	0.4316	y = 1.7871$e^{-5\times 10^{-4}x}$	0.0067
BF	y = 1.4203e^0.0067x^	0.7466	y = 2.3278e^0.0007x^	0.0338
ST	y = 0.5349e^0.0063x^	0.6431	y = 0.8766e^0.0003x^	0.0043
IS	y = 0.4640e^0.0044x^	0.3564	y = 0.7605e^−0.002x^	0.0653
SS	y = 0.1905e^0.0074x^	0.3213	y = 0.3123e^0.0014x^	0.0170
PM	y = 0.5438e^0.0033x^	0.1434	y = 0.8913e^−0.003x^	0.1011
Total	y = 4.494e^0.0079x^	0.7861	y = 7.3654e^0.0019x^	0.1685

y = muscle yield or proportion; x = cold carcass weight.

## Data Availability

The data presented in this study are available on request from the corresponding author.

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
