# Peer review of "A Descriptive Study on the Carcass, Muscle, and Offal Yields of the Plains Zebra (Equus quagga) Harvested in Two Seasons"

_animals, 2022, doi:10.3390/ani13010050_

Round 1

Reviewer 1 Report

General Comments

The manuscript entitled "A Descriptive Study on The Carcass, Muscle, and Offal Yields of The Plains Zebra (Equus quagga) Harvested in Two Seasons" is well written and does have some novelty with respect to the use of alternative feed resources as an income generator and an alleviator of food insecurity. I believe that the manuscript should be published with minor corrections. 

Methodology

1. The author could have included a map of the sample sites to give readers a graphical appreciation for their space.

2. Line 175 and 177: Should it be a '+' or '- 'sign? Please revise.

3. If inferential statistics were performed on the carcass parameter and body weight it would have made the work more vigorous. As it is with descriptive analysis only numerical differences can be seen. A follow up study can be done in the future to analysis various parameter such as season, age and sex on carcass parameters and also nutritive and sensory analysis of zebra meat. (This is just a suggestion)

Results and Discussion

There are many 'errors with references not found'. Please revise this to include the appropriate references.

Conclusion

This section should be re written to be more concise. A conclusion should state the salient findings of the manuscript and not read as a discussion.

Author Response

General Comments

The manuscript entitled "A Descriptive Study on The Carcass, Muscle, and Offal Yields of The Plains Zebra (Equus quagga) Harvested in Two Seasons" is well written and does have some novelty with respect to the use of alternative feed resources as an income generator and an alleviator of food insecurity. I believe that the manuscript should be published with minor corrections. Thanks-you for your kind and insightful comments. We have attempted to address all the points you raised.

Methodology

1. The author could have included a map of the sample sites to give readers a graphical appreciation for their space. In the end, we decided not to include a map as these two regions where the zebra were samples are on the extreme southern part of their range.

2. Line 175 and 177: Should it be a '+' or '- 'sign? Please revise. The ± (plus/minus) is commonly used to indicate that the temperature fluctuated around 4°C – this fluctuation is caused by the chilling unit going into its thawing cycle.

3. If inferential statistics were performed on the carcass parameter and body weight it would have made the work more vigorous. As it is with descriptive analysis only numerical differences can be seen. A follow up study can be done in the future to analysis various parameter such as season, age and sex on carcass parameters and also nutritive and sensory analysis of zebra meat. (This is just a suggestion) I appreciate this suggestion, and have highlighted this aspect in the Conclusion (which we have rewritten as suggested by yourself and another Reviewer.

Results and Discussion

There are many 'errors with references not found'. Please revise this to include the appropriate references. Thank-you for highlighting this error – it was also mentioned by the editor. It seems that with the system transforming the document, some of the hyperlinks to the Figures/Tables and References were broken.

Conclusion

This section should be re written to be more concise. A conclusion should state the salient findings of the manuscript and not read as a discussion. We have rephrased the Conclusion to be more precise; note we also included the section as suggested above about the possible focus of future research.

Reviewer 2 Report

The manuscript is interesting and brings some new scientific knowledge.

 I have several suggestions and proposals for the manuscript

1It is necessary to clearly describe the term „farm“, whether it is a farm where animals have a limited radius of movement, whether there is human feeding, whether there are predators, etc.

2Check with the statistics editor the way certain data are displayed in the table (e.g. non-uniformity in the display of the same data, etc.)

3In the introductory part, the issue of zebra meat production is described. It would be interesting and useful to make a regression model of the proportion of meat in the gross mass per slaughter for different weight/age categories

4The research was conducted in the southern hemisphere. Perhaps it would be easier to follow the text if instead of summer/winter it was written the colder/warmer part of the season.

5There are technical errors in several places in the manuscript

TThe main failure of manuscript - The research was conducted on a small number of samples (m/f; age/weight class). An additional problem is not knowing the age of some of the sampled animals, which can have a significant impact on the presented results. The observed differences in the hunting season (winter; summer) may be the result of different ages. A comparison sample would be correct if comparing two groups of animals of the same age/sex, shooted in two different seasons.

Author Response

The manuscript is interesting and brings some new scientific knowledge.

 I have several suggestions and proposals for the manuscript Thanks you for your king remarks and suggestions. We have addressed these as far as possible.

1It is necessary to clearly describe the term „farm“, whether it is a farm where animals have a limited radius of movement, whether there is human feeding, whether there are predators, etc. This has been addressed in lines 131-133; 152-153, briefly there were no natural predators and the animals were kept in (and out) by means of an exclusion fence.

2Check with the statistics editor the way certain data are displayed in the table (e.g. non-uniformity in the display of the same data, etc.) Thank-you for this comment, the statistical analyses and the compilation of the Tables/figures were done in consultation with a statistician Prof Martin Kidd [mkidd@sun.ac.za]

3In the introductory part, the issue of zebra meat production is described. It would be interesting and useful to make a regression model of the proportion of meat in the gross mass per slaughter for different weight/age categories I agree that this would be interesting, however, the zebra were either all mature or close to their mature weight, the muscles removed and weighed did not make up a high proportion of the total meat, for this we would need to do a thorough deboning and weigh the meat:bone ratio. None the less this is an excellent idea for future research.

4The research was conducted in the southern hemisphere. Perhaps it would be easier to follow the text if instead of summer/winter it was written the colder/warmer part of the season. This is a valid point; we have included these two terms in parenthesis when we describe the ‘seasons’.

5There are technical errors in several places in the manuscript

The main failure of manuscript - The research was conducted on a small number of samples (m/f; age/weight class). An additional problem is not knowing the age of some of the sampled animals, which can have a significant impact on the presented results. The observed differences in the hunting season (winter; summer) may be the result of different ages. A comparison sample would be correct if comparing two groups of animals of the same age/sex, shooted in two different seasons. We agree 100% that the results were confounded by numerous factors. None the less, we still believe that this ‘exploratory’ findings are worthy of being published, as there are numerous zebra being culled and their meat exported. These factors that could influence the results and the need for research to investigate their effects/influences has been highlighted in the Conclusion section.

Reviewer 3 Report

The manuscript presents baseline data an meat yield of culled zebra. Animals had been culled in 2 regions (one in winter, one in summer) in South Africa. Since differences were found between summer- and winter-harvested specimens, it would be interesting to address if geographical differences in addition to seasonal differences may have contributed to different carcass weights etc (the two different locations are mentioned in the abstract, and in detail in the manuscript body, but not reflected in the title). The data are of high relevance since they prove that this type of wild horse, adapted to harsh environments, can be utilized as a sustainable meat source for the domestic market as well as for export.

There are some details that should be cleared/fixed before publishing this very interesting manuscript:

lines 168-172: unclear if carcasses have been skinned before quartering / when skin was removed.

lines 212, 229,245, 283, 465,553,573: there is an error mark, is this the reference to Tabs./Figs.?

lines 297, 317, 327 and others "biceps"

line 232 (and others), for the r or R^2 values, it would be interesting if they were stat. significant, even when weak.

line 311: not clear; does it mean that the percentages (0.4-0.6) were not significantly different or that the contribution of the three muscles to the carcass weight was negligible?

Author Response

The manuscript presents baseline data on meat yield of culled zebra. Animals had been culled in 2 regions (one in winter, one in summer) in South Africa. Since differences were found between summer- and winter-harvested specimens, it would be interesting to address if geographical differences in addition to seasonal differences may have contributed to different carcass weights etc (the two different locations are mentioned in the abstract, and in detail in the manuscript body, but not reflected in the title). The data are of high relevance since they prove that this type of wild horse, adapted to harsh environments, can be utilized as a sustainable meat source for the domestic market as well as for export.

There are some details that should be cleared/fixed before publishing this very interesting manuscript:

lines 168-172: unclear if carcasses have been skinned before quartering / when skin was removed. The animals had been skinned before quartering – we have rephrased this so that it is clearer (thankyou for raising this point). Inserted in Lines 171-172.

lines 212, 229,245, 283, 465,553,573: there is an error mark, is this the reference to Tabs./Figs.? Thank-you for highlighting these, it seems that with the transformation of the manuscript, the hyperlinks have been corrupted.

lines 297, 317, 327 and others "biceps" The ‘s’ has been added throughout the document.

line 232 (and others), for the r or R^2 values, it would be interesting if they were stat. significant, even when weak. As they were weak (all were significant), we felt that we did not want to emphasise their importance as correlations/predictive equations, particularly with such a small sample size could lead to wrong deductions being made by the readers.

line 311: not clear; does it mean that the percentages (0.4-0.6) were not significantly different or that the contribution of the three muscles to the carcass weight was negligible? Now Line 314 has now been rephrased to indicate that it is each individual muscle’s contribution to the total carcass weight that these % are referring to.